

# Unraveling the structure and composition of Varadero Reef, an improbable and imperiled coral reef in the Colombian Caribbean

Valeria Pizarro[1], Sara C. Rodríguez[2], Mateo López-Victoria[2], Fernando A. Zapata[3], Sven Zea[4], Claudia T. Galindo-Martínez[5], Roberto Iglesias-Prieto[5], Joseph Pollock[5] and Mónica Medina[5]

[1] Ecomares NGO, Cali, Valle, Colombia
[2] Department of Natural Sciences and Mathematics, Pontifica Universidad Javeriana, Cali, Valle, Colombia
[3] Department of Biology, Universidad del Valle, Cali, Valle, Colombia
[4] Centro de Estudios en Ciencias del Mar—CECIMAR, Universidad Nacional de Colombia—Sede Caribe, Santa Marta, Magdalena, Colombia
[5] Department of Biology, Pennsylvania State University, State College, PA, United States of America

Corresponding author
Valeria Pizarro,
valeria.pizarro@ecomares.org,
valeria.santamarta@gmail.com

## ABSTRACT

Coral reefs are commonly associated with oligotrophic, well-illuminated waters. In 2013, a healthy coral reef was discovered in one of the least expected places within the Colombian Caribbean: at the entrance of Cartagena Bay, a highly-polluted system that receives industrial and sewage waste, as well as high sediment and freshwater loads from an outlet of the Magdalena River (the longest and most populated river basin in Colombia). Here we provide the first characterization of Varadero Reef's geomorphology and biological diversity. We also compare these characteristics with those of a nearby reference reef, Barú Reef, located in an area much less influenced by the described polluted system. Below the murky waters, we found high coral cover of 45.1% (±3.9; up to 80% in some sectors), high species diversity, including 42 species of scleractinian coral, 38 of sponge, three of lobster, and eight of sea urchin; a fish community composed of 61 species belonging to 24 families, and the typical zonation of a Caribbean fringing reef. All attributes found correspond to a reef that, according to current standards should be considered in "good condition". Current plans to dredge part of Varadero threaten the survival of this reef. There is, therefore, an urgent need to describe the location and characteristics of Varadero as a first step towards gaining acknowledgement of its existence and garnering inherent legal and environmental protections.

## INTRODUCTION

Coral reefs provide important ecosystem services (*Moberg & Folke, 1999*), but many currently face unprecedented pressure from multiple natural and anthropogenic stressors (*Wilkinson, 2008*). Caribbean reefs have been particularly impacted, with coral cover

decreasing from an average of 50% to 10% in just four decades (*Jackson et al., 2014*). Coral cover loss has resulted in a phase shift from coral to macroalgal domination with a concurrent increase in sponge abundance (e.g., *Rose & Risk, 1985*; *Szmant, 2002*; *Ward-Paige et al., 2005*; *Chavez-Fonnegra, Zea & Gomez, 2007*; *Malaio, Turingan & Lin, 2008*; *Jackson et al., 2014*).

Coral reef ecosystems, built mainly by scleractinian corals, typically thrive within a narrow range of environmental conditions characterized by low sedimentation rates, low nutrient availability (i.e., oligotrophic waters), high light penetration, warm waters (e.g., around 28 °C) and salinity between 33 and 36 psu (*Kleypas, McManus & Meñez, 1999*; *Díaz et al., 2000*; *Sheppard, Davy & Pilling, 2009*). Although reefs can be found outside these ranges in "extreme" environmental conditions, such reefs are typically dominated by a low number of resistant specialist species. Some examples include reefs under higher water temperatures in the Persian Gulf and Hawaii (*Oliver & Palumbi, 2009*; *Riegl & Purkis, 2012*), reefs under low pH waters in Japan and Papua New Guinea (*Fabricius et al., 2011*; *Inoue et al., 2013*), and reefs under high salinity such as those at the Arabian Sea where salinity can exceed 45 psu and temperatures regularly top 34 °C (*Rezai et al., 2004*).

In 2013, a reef was discovered under unexpected conditions below a thick layer of highly turbid water at the mouth of Cartagena Bay, Colombia (*López-Victoria, Rodríguez-Moreno & Zapata, 2015*). This reef, known as Varadero, is located south of Tierra Bomba Island, at the mouth of the highly polluted Bay. The man-made "Canal del Dique" dumps industrial and sewage waste as well as discharges of sediment from the Magdalena River into the vicinity of Varadero. With a drainage basin covering 24% of Colombia's surface area (27.3 million hectares), the Magdalena River feeds approximately $144 \times 10^6$ tons of suspended solids into Cartagena Bay each year. This enormous sediment load has contributed to the demise of the Bay's once vibrant coral reefs (*Restrepo et al., 2006*). Paradoxically, Varadero Reef has not only survived, but thrived with up to 80% coral cover dominated by large *Orbicella* spp. colonies, the major reef-building corals in the Caribbean (*López-Victoria, Rodríguez-Moreno & Zapata, 2015*).

Despite its close proximity to the city of Cartagena, Colombia (>1 million inhabitants), Varadero Reef remained concealed due to the perception that local environmental conditions were incompatible with reef growth. High levels of sedimentation and turbidity have previously been shown to drive coral bleaching and disease that can ultimately lead to coral death (*Bruno et al., 2003*; *Harvell et al., 2007*; *Pollock et al., 2014*). Here we provide a preliminary characterization of Varadero Reef, including its geomorphology (i.e., size, shape and location) and biological diversity (i.e., coral, fish and sponge community composition). We also compare these characteristics with those of a nearby reference reef, Barú Reef, located 4.5 km south of Varadero, in a location much less influenced by runoff from the Canal del Dique and the city of Cartagena.

Current plans to dredge part of Varadero threaten the survival of this reef and could hinder researchers' ability to gain insights into the factors that have allowed corals to thrive under such unusual conditions. There is, therefore, an urgent need to describe the location and characteristics of Varadero as a first step towards gaining acknowledgement of its existence and garnering inherent legal and environmental protections.

## MATERIALS & METHODS

In order to supplement the brief, general description of Varadero Reef reported by *López-Victoria, Rodríguez-Moreno & Zapata (2015)*, detailed geomorphological and biological surveys were performed between 2014 (March) and 2015 (March and October). During the March 2015 field trip, the reef's geographic extent was assessed by two researchers diving along the reef edge with a GPS, recording in tracking mode, attached to an accompanying buoy. Data from the GPS was downloaded and analyzed using the GIS software Garmin BaseCamp, from which a detailed map of the reef was subsequently produced. The reef's coral diversity was characterized by two coral experts performing three replicate profiles starting in the deepest zone (in direction to open sea) towards Cartagena Bay (shallowest zone). These annotations, including coral community composition at multiple depths, were analyzed as in *Geister (1977)*. All profiles were compared and compiled to obtain a detailed profile of the reef's coral community structure and diversity.

The vertical attenuation coefficients ($K_d$) were determined at both sites using the cosine corrected sensor of a diving pulse modulated fluorometer (PAM) (Waltz, Germany). The PAM sensor was calibrated against a traceable reference sensor LiCor (USA). A diver operating the PAM maintained the instrument in a horizontal position and triggering the data collection system of the fluorometer at different depths. The maximum excitation pressure over photosystem II ($Q_m$) was calculated in both sites using the effective quantum yield of photosystem II at apparent noon ($\Delta F/Fm'$) and the maximum quantum yield of charge separation at dusk ($Fv/Fm$) (*Iglesias-Prieto et al., 2004*).

A detailed benthic community assessment was also conducted to evaluate sessile and mobile species composition, fish diversity and abundance, and sponge richness. To allow comparison of Varadero with a nearby reef that reflected typical Caribbean reef environmental conditions, a reef on the Barú Peninsular (from now on Barú Reef) was also surveyed. At each reef, five stations were established and two 30-m transects were deployed in the same landscape unit (i.e., reef type and depth). Quadrats (50 by 50 cm) were placed every three meters on each side of the transect and photographed for a total of 20 photo quadrats per transect. Each photograph was analyzed using Coral Point Count 4.1 software (*Kohler & Gill, 2006*), which randomly places 50 points within the quadrat for a randomly stratified methodology (*Kohler & Gill, 2006; Dumas et al., 2009; Andersen et al., 2012*). The benthic component below each point was identified and categorized as coral (identified to species level), sponge (identified to species level), algal overgrown dead coral, sand/rubble or other invertebrates (e.g., tunicates, gorgonians or zoantharians). Mobile reef invertebrates were also assessed using the same benthic transects. A visual census was preformed of all sea urchins, conchs, and lobsters within a 1-m wide band of the transect. Macroalgal communities were characterized by randomly selecting five photo quadrats per transect, randomly placing 10 points within each quadrat (using Coral Point Count 4.1), and categorizing any observed macroalgae as fleshy, coralline or turf. Fleshy algae were identified to genus level. To compare Varadero and Barú Reefs, species richness, abundance and composition were tested for normal distributions (Shapiro–Wilk's test)

then compared using a two sample Student's *t*-test in the software PAST version 3.14 (*Hammer, Harper & Ryan, 2001*).

During exploratory dives, sponges were visually identified while swimming over the reef. Photographs and small samples were also taken for downstream spicule examination in cases when sponges could not be readily distinguished in the field. Species lists were made for both Varadero and Barú Reefs, separately for the upper terrace (down to 10–13 m) and slope (below 10–13 m) zones. Sponge species present within each of the $30 \times 2$-m$^2$ band transects in the shallow terrace zone of Varadero ($n = 7$ transects) and Barú ($n = 4$ transects) were also recorded. This sampling scheme permitted calculation of gross abundances as percent frequency of occurrence (number of transects in which a sponge was present/total transects) and species richness per transect. Data on total coral and sponge cover obtained in 10 photo transects (covering 5 m$^2$ each, see above) in the upper terrace of each locality were also analyzed for trends in cover of sponges *vs.* corals *vs.* available substratum using simple correlation analysis. For sponge identification in the laboratory, small fragments of each collected sponge were digested in commercial bleach to obtain free spicules, which were observed under a light microscope. Species were identified using specialized literature and extensive local knowledge/experience (see *Zea, 1987*; *Zea, Henkel & Pawlik, 2014*).

Overall fish diversity and community composition were visually assessed. In order to compile fish species lists for each reef, a team of three divers recorded all fishes observed while exploring the general reef areas of Varadero and Barú during a total of 8 dives on each reef (approximately 1-hour per dive), in 2014 and 2015. In 2015, 22 visual censuses were performed along $30 \times 2$-m$^2$ belt transects ($n = 15$ at Varadero and $n = 7$ at Barú) to characterize fish community composition. All individuals observed within each belt transect were counted and these counts were used to estimate mean species richness, diversity (Shannon's H'), dominance (Simpson's D) and evenness (Pielou's J'). These community variables were compared between Varadero and Barú using a two sample Student's *t*-test, after establishing that the data met assumptions of normality and homoscedasticity with Shapiro–Wilk's and *F* tests, respectively. All tests were performed using PAST 3.14 (*Hammer, Harper & Ryan, 2001*).

To assess species abundance differences between sites, a regression analysis of mean species abundance was performed along with paired Student's *t*-tests. Given the different sampling efforts between the two localities, a sample-based rarefaction procedure was carried out to compare fish species richness between Varadero and Barú. Finally, a non-Metric Multidimensional Scaling (nMDS) analysis was carried out using Jaccard's similarity index (based on species occurrence) and the Bray–Curtis similarity index (based on the log $(x + 1)$ transformed abundance data) to examine differences in assemblage structure between the two localities based on species composition and abundance, respectively. The nMDS analysis was complemented with analyses of similarity (ANOSIM) based on either Jaccard or Bray-Curtis similarity. All statistical analyses and calculation of community indices were performed using the software PAST 3.11 (*Hammer, Harper & Ryan, 2001*).

## RESULTS

### Geomorphology and optical properties

Located between the Bocachica navigation channel and the island of Barú, Varadero Reef has an area of approximately 1.12 km$^2$ (Fig. 1). The reef has two contrasting zones, the first (0.44 km$^2$) is a well-developed reef where scleractinian coral colonies dominate the substratum. The second (0.68 km$^2$) is a carbonated terrace with scattered corals, octocorals, a few other benthic species and sand patches with seagrasses (Figs. 1C, 2). The largest seagrass beds were observed near the islands of Draga and Abanico (Fig. 1C). Analyses of the vertical attenuation coefficients of the water in both sites indicate significant vertical stratification. We identify an upper layer with high attenuation values located between the surface and 3–5 m depth. Comparisons between the attenuation coefficients of the first layer at both sampling sites indicate significantly ($p < 0.001$ ANOVA) higher attenuation values for Varadero Reef ($0.336 \pm 0.050$ m$^{-1}$, average $\pm$ SE, $n = 32$) relative to Barú Reef ($0.243 \pm 0.053$ m$^{-1}$, $n = 11$). In some cases, we identify a second layer with $K_d$ values ranging between 0.193 and 0.051 m$^{-1}$ at depths above the limit of the first layer between three to five meters (Fig. 3). Depending on the depth profile of the reef, some corals were completely contained within the first optical layer (Figs. 2 and 3). We recorded the maximum excitation pressure of photosystem II for *Orbicella faveolata* colonies growing in the shallow parts of both reefs. In both cases corals were exposed to irradiances high enough to induce significant levels of photoprotection at noon with $Qm$ values of $0.208 \pm 0.109$, average $\pm$ SE, $n = 25$ at 4.5 m depth and $0.249 \pm 0.052$, $n = 25$ at 6.0 m depth for Varadero and Barú Reefs respectively.

### Coral and benthic community

In total, 42 scleractinian coral and four hydrocoral species (Families Milleporidae and Stylasteridae) were identified at Varadero (Table S1). These species include several threatened species such as the acroporids (*Acropora cervicornis* and *A. palmata*). Depth profiles indicate that Varadero Reef's calcareous matrix starts at around 27 to 35 m depth (Fig. 2). At greater depths, moving towards open sea, the sand bottom has small patches of sponges and black corals (Anthipatharia). Coral cover from 27 to 35 m until approximately 10 to 12 m is relatively low (1 to 5%) and the reef slope is around 45°. Coral communities at this depth range are dominated by *Agaricia* spp. (*A. lamarcki*, *A. grahamae*), *Madracis* spp. and *Helioseris cucullata*. At 25 m and shallower, small plate-like growth forms of *Siderastrea siderea*, *Montastraea cavernosa* and *Mycetophyllia aliciae* were observed. Besides corals, tube and branching sponges, encrusting algae and cyanobacteria are present. At 18 m and shallower, small patches of *Agaricia tenuifolia* start to appear, becoming more abundant until they dominate the landscape between 12 and 10 m. Between 12 and 10 m, live coral cover is 40–45%, the slope decreases to 25–30° and other scleractinian species are present, including *Colpophyllia natans*, *H. cucullata*, *Madracis auretenra*, *O. faveolata*, *Porites astreoides*, and *Scolymia cubensis* become more common. At 10–12 m depth, growth morphologies of typically massive species are plate-like and small (~10–40 cm maximum diameter). *Madracis auretenra* also forms scattered monospecific patches in this area.

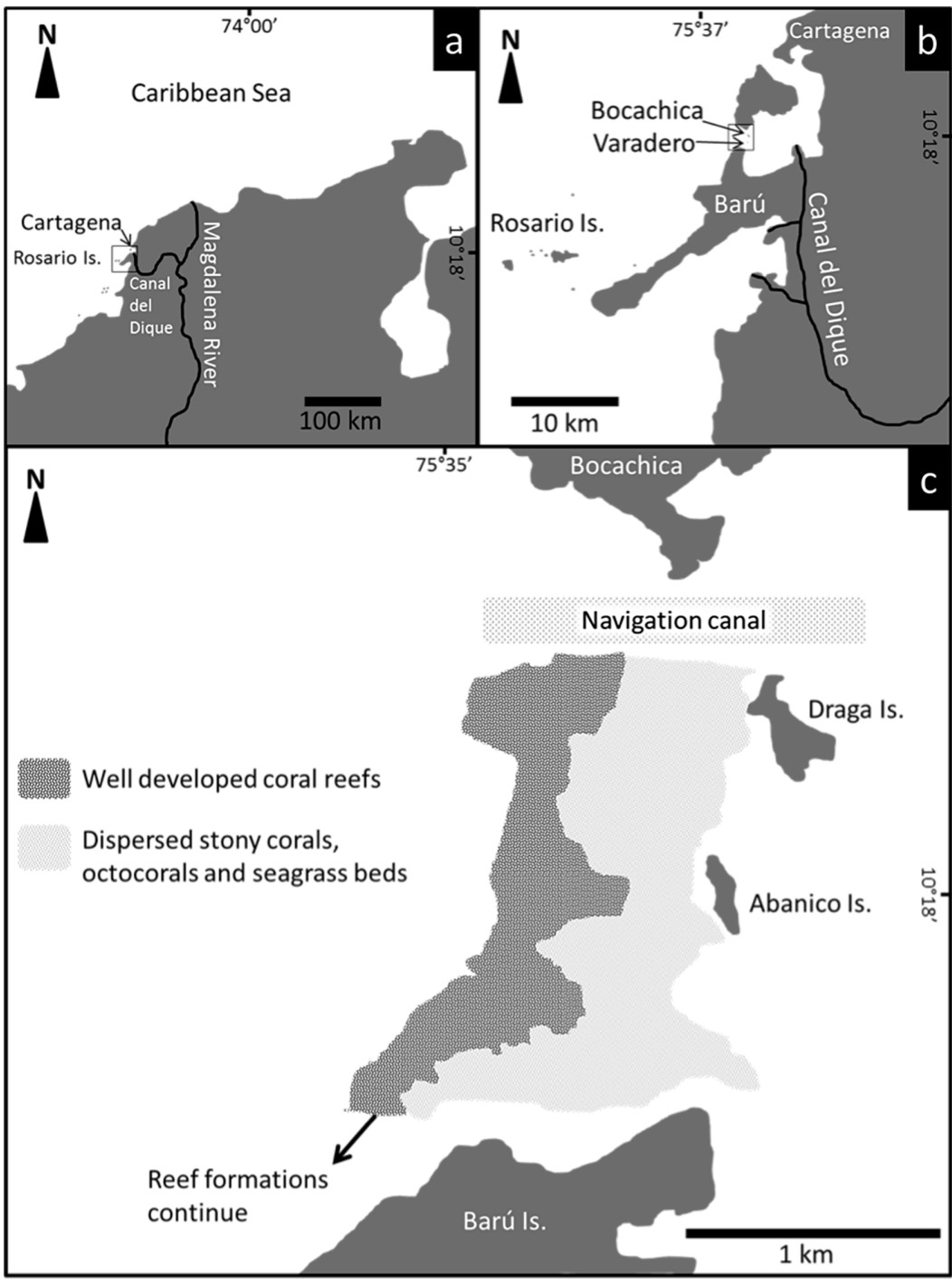

**Figure 1    Location and distribution of Varadero Reef.** The reef continues to the South towards Barú Island.

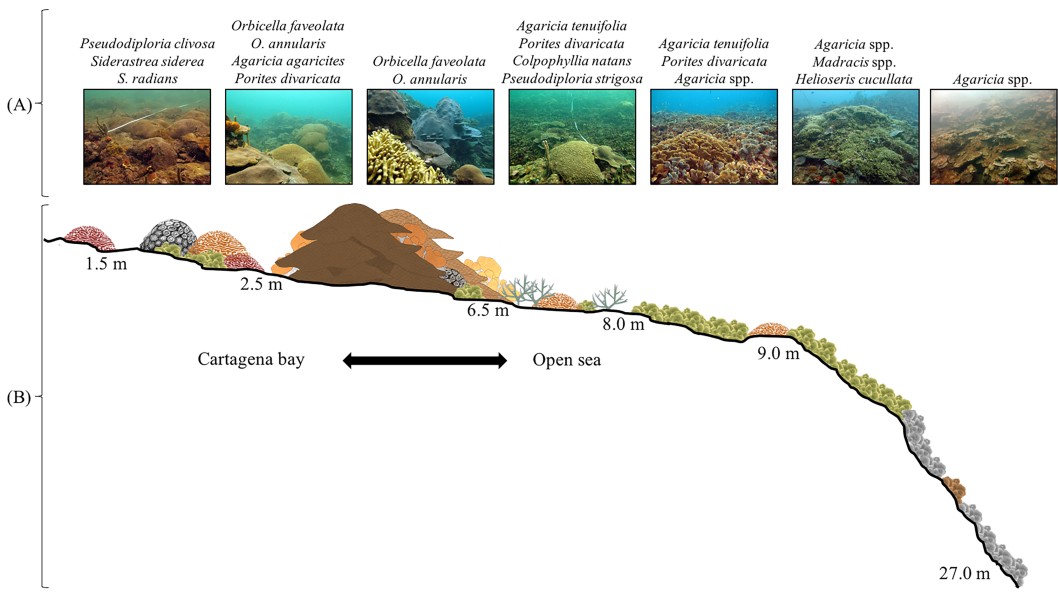

**Figure 2 Varadero Reef profile.** Profile of Varadero Reef showing the typical zonation and coral composition (A and B). Photographs in (A) correspond to each sector of the reef and the dominant scleractinian coral taxon (Credit: coauthors).

At approximately 8 m depth, the slope decreases to 10–15°, corals are more abundant and larger (up to 2–3 m diameter), but the main coral matrix is still dominated by *Agaricia tenuifolia* and in some areas is mixed with *Porites divaricata*. The morphology of typically massive coral species is a mix of massive and plate. The most common species are *Meandrina meandrites*, *Montastraea cavernosa*, *Mycetophyllia ferox, Orbicella annularis*, *O. faveolata*, *Pseudodiploria strigosa,* and *Siderastrea siderea*. At this depth, it is possible to find *A. cervicornis*. At 6 m, coral cover increases to 50–60%, massive corals become dominant (especially *Orbicella* spp.), and patches of *Agaricia tenuifolia* and *Porites divaricata* can be found in sand patches. Between 5 and 3 m, massive corals dominate the reefscape, *Orbicella annularis* and *O. faveolata* colonies with diameters exceeding 5 m are common and the slope decreases to almost 0°. Other common coral species include *Agaricia. agaricites*, *A. tenuifolia, Colpophyllia natans, Millepora alcicornis, M. complanata, M. striata, Mycetophyllia aliciae, Pseudodiploria strigosa,* *Porites astreoides*, *P. divaricata, Scolymia cubensis* and *Siderastrea siderea*. Live coral cover is higher than 50% and colonies of *Acropora cervicornis, A. palmata* and *A. prolifera* are found scattered throughout the reef. This area of high coral cover which is dominated by large colonies of *Orbicella* spp. continues until around 3 m. At this depth, coral colony size and abundance decreases. Common coral species, between 3 and 2 m depth include *Agaricia fragilis, A. tenuifolia, Favia fragum, Orbicella faveolata, Pseudodiploria clivosa, P. strigosa, Porites astreoides, P. divaricata, Montastraea cavernosa, Siderastrea siderea*, as well as the milleporids *Millepora complanata* and *M. striata*. Most of the massive coral species' growth morphologies change to crustose, and the reef slope is less than 10°. Calcareous terraces appear at 2 m. In this area, dispersed corals (*Pseudodiploria clivosa, Siderastrea radians* and *S. siderea*), octocorals,
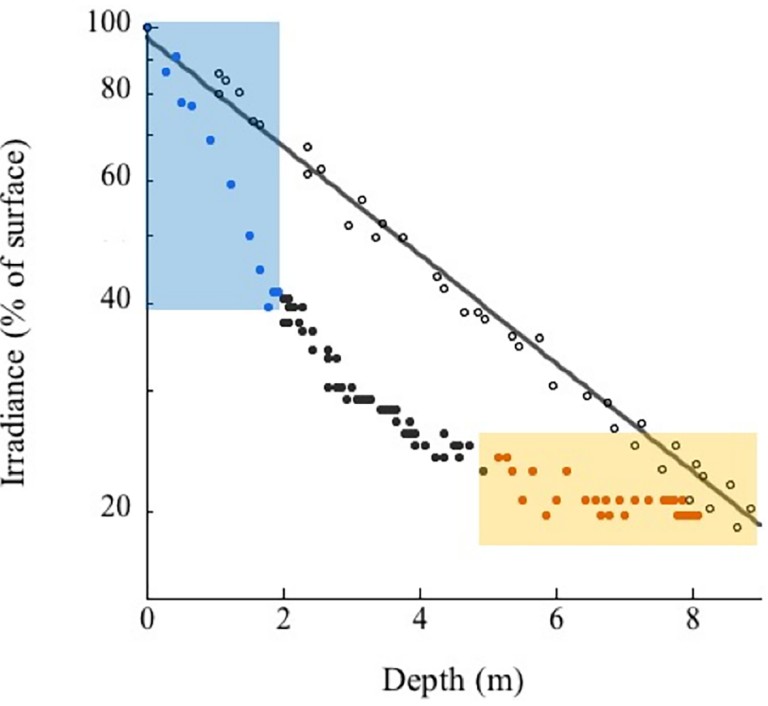

**Figure 3 Varadero Reef optical properties.** Analyses of the variations in the optical properties of the water column in Varadero Reef (solid circles) indicate the presence of highly stratified water masses. The blue symbols in the blue shaded area highlight the upper layer with $K_d$ values of 0.488 m$^{-1}$, the black symbols indicate transition region with $K_d$ of 0.19 m$^{-1}$ whereas the orange symbols in the shaded area indicate the presence of very clear waters with $K_d$ values of 0.041. For comparison the monotonic vertical attenuation for the Rosario Island is presented (open circles) with $K_d$ values of 0.165 m$^{-1}$.

and sand patches are common. Towards the Bay, close to the islands of Abanico and Draga, seagrasses (i.e., *Thalassia testudinum* and *Halodule wrightii*) are common.

Varadero Reef's benthos between 3 and 15 m is dominated by live coral (45.1 ± 3.9%) and algae-overgrown dead coral (47.5 ± 4.0%; average ± SE). Sand and rubble (4.6 ± 0.6%), sponges (0.7 ± 0.1%) and other invertebrates (gorgonians, zoantharians, etc.) (1.8 ± 0.9%) were also observed. In total, 38 coral species (scleractinian and fire corals) were identified at this depth. The most abundant species are *Orbicella faveolata* (38.1%), *Agarcia agaricites* (28.8%), *O. annularis* (14.4%) and *A. tenuifolia* (12.2%) (Table S1). Similar to Varadero, the most common benthic components at Barú Reef are algae-overgrown dead coral (56.9 ± 2.7%) and live coral (38.1 ± 3.2%). The other benthic categories assessed show low percentage cover of sand and rubble (3.4 ±1.6%), sponges (0.8 ± 0.2%) and other invertebrates (0.9 ± 0.3%). In total, 35 coral species were identified, and, similar to Varadero, the most common were *Orbicella faveolata* (25.6%), *Agaricia agaricites* (11.3%), *O. annularis* (10.4%) and *A. tenuifolia* (4.5%) (Table S1).

## Sponge community

In total, at Varadero and Barú fifty sponge species were observed with 38 and 31 species at each reef, respectively. Survey transects at upper shallow terraces (between 3 and 10 m

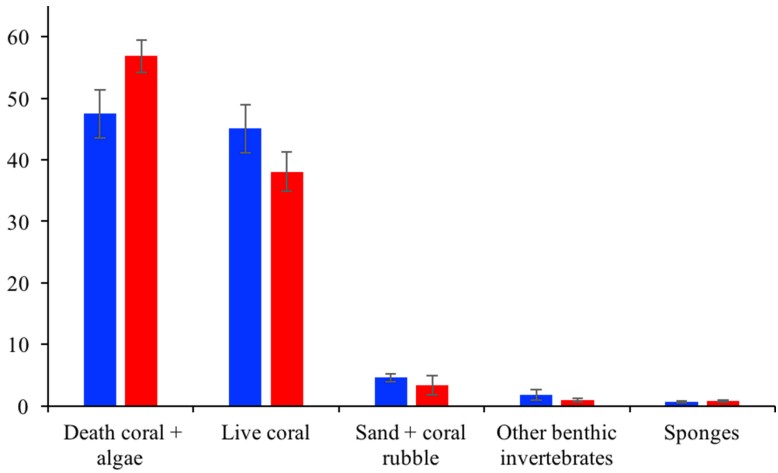

**Figure 4 Varadero and Barú benthic cover.** Average benthic coverage Varadero (blue) and Barú (red) Reefs. Error bars indicate standard error.

depth) at Varadero Reef showed higher sponge species richness (36 in total) than that of upper shallow terraces in Baru Reef (25 in total) although the number of species per transect were not significantly different ($t$-Student test, $p = 0.86$), 10.0 $\pm$1.23 species per transect (mean $\pm$ 1 standard error, $n = 7$ transects) for Varadero, and 10.5 $\pm$ 2.36 for Barú ($n = 4$ transects) (Table S2). Eight species, arranged by abundance, were observed in greater than 50% of terrace transects on both reefs, *Mycale laevis, Niphates erecta, Ircinia felix, Monanchora arbuscula, Lissodendoryx colombiensis, Haliclona wallentinae, Cliona laticavicola* and *Scopalina ruetzleri*. None of these common species were exclusive to either reef, and when reef-specific species were observed, they were typically comprised of single occurrences. Visually, sponge abundance was similarly low in both Varadero and Barú Reef terraces though there were sponge patches growing on dead coral. Mean coral cover estimated from phototransects was slightly but not significantly higher in Varadero than in Barú (45.1 $\pm$ 14.3% vs. 38.1 $\pm$ 12.0% respectively, $t$-Student test, $p = 0.18$, $n = 10$ transects per site, Fig. 4). Sponge cover was equally low and similar between the two localities (0.66 $\pm$ 0.21% and 0.80 $\pm$ 0.25% respectively, $t$-Student test, $p = 0.52$). Moreover, correlations between per-transect total coral and sponge cover, although negative as expected, were not significant (Varadero, $r = -0.42$, $p = 0.22$; Barú, $r = -0.06$, $p = 0.86$). Mean sponge cover was also not significantly correlated with the availability of dead coral substratum (covered with turf and macroalgae, Varadero, $r = 0.42$, $p = 0.23$; Barú, $r = 0.36$, $p = 0.30$), which was higher in Barú (56.9 $\pm$ 18.0%) than in Varadero (51.4 $\pm$ 16.3%).

## Fish community

A total of 61 fish species from 24 families was observed at Varadero Reef compared to 44 species from 22 families observed at Barú. While a total of 67 species were observed at both sites combined, 38 species were common to both. Twenty-four species were observed at Varadero only, while six species were observed exclusively at Barú. Overall, Jaccard's

**Table 1 Fish assemblage at Varadero and Barú Reefs.** Fish assemblage attributes estimated through visual censuses on 30 × 2-m² belt transects made at Varadero and Barú Reefs.

| Community attribute | Varadero ($n = 15$) | | Barú ($n = 7$) | | $t$ | $p$ |
|---|---|---|---|---|---|---|
| | Mean | ±SD | Mean | ±SD | | |
| Species richness | 12.4 | 3.0 | 15.0 | 2.4 | −1.99 | 0,06 |
| Number of individuals | 55.6 | 15.9 | 74.1 | 14.4 | −2.62 | 0,02 |
| Dominance (Simpson's D) | 0.18 | 0.05 | 0.16 | 0.04 | 0.94 | 0,36 |
| Diversity (Shannon's H') | 2.0 | 0.3 | 2.2 | 0.2 | −1.36 | 0,19 |
| Evenness (Pielou's J') | 0.81 | 0.07 | 0.80 | 0.04 | 0.27 | 0,79 |

coefficient of similarity considering the full fish species list of each site was 0.57 (Table S3). The number of species per family was similar between Varadero and Barú ($r = 0.90$, $p \ll 0.001$, $n = 26$ families) and at both sites damselfishes (Pomacentridae) were the most species rich (eight and seven species at Varadero and Barú, respectively), followed by wrasses (Labridae; five species at each site), groupers (Serranidae; five and four species, respectively) and parrotfishes (Scaridae; four species at each site; Table S3).

Considering only data from visual censuses, a total of 834 individuals belonging to 36 species were observed at Varadero, while only 519 individuals of 32 species were observed at Barú. Correcting for differences in sampling effort, sample-based rarefaction indicated that, for the same number of samples, species richness was slightly greater at Barú than at Varadero (Figure S1). Nonetheless, mean species richness within transects at Varadero did not differ significantly from mean species richness at Barú (Table 1). Except for the total number of individuals per transect, which was on average significantly greater at Barú than at Varadero, none of the other community parameters (Simpson's Dominance D, Shannon's Diversity H', and Pielou's Evenness J') differ significantly between Varadero and Barú ($p > 0.05$) (Table 1). Even though there was a highly significant positive correlation between the abundance of species common to both sites (considering only species observed in transects at both sites; $r = 0.95$, $p \ll 0.001$, $n = 26$ species), a paired Student's $t$-test indicated that mean abundance was significantly greater at Barú than at Varadero (mean difference $= 0.78$, $t = -2.51$, $p = 0.019$).

Results of the nMDS analysis showed that there was a great deal of overlap in fish assemblage structure between Varadero and Barú considering either species composition alone (based on Jaccard's similarity; Fig. 5A) or species abundance and composition (based on Bray Curtis's similarity; Fig. 5B). ANOSIMs based on these two similarity measures indicated that the fish assemblage at Varadero did not differ significantly from that at Barú (Jaccard-based ANOSIM, $R = 0.03$, $p = 0.37$; Bray–Curtis-based ANOSIM, $R = -0.06$, $p = 0.69$).

# DISCUSSION

Caribbean coral reefs are declining rapidly due to anthropogenic activities (e.g., overfishing, pollution, etc.), climate change and the synergies between these factors. Caribbean reefs have experienced declines in coral cover (and increases in macroalgae, cyanobacterial mats and

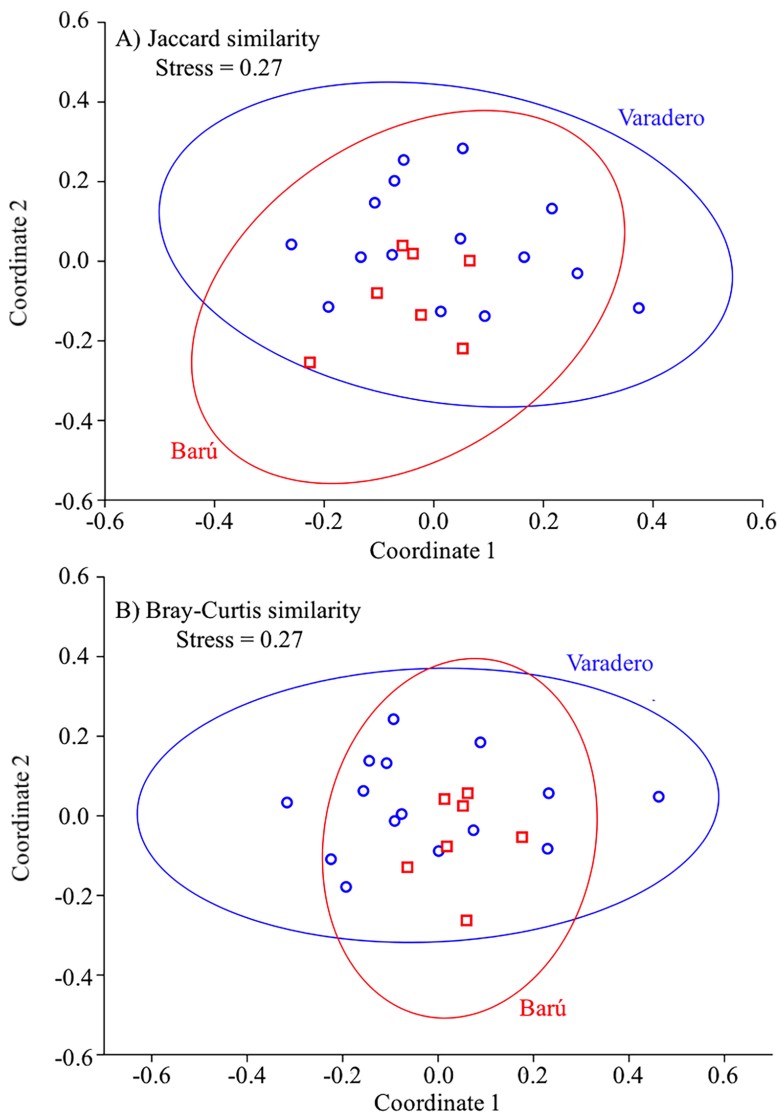

**Figure 5** **Fish presence-absence and abundance data for Varadero and Barú Reefs.** Non-metric multidimensional scaling analysis biplots based on (A) presence-absence data (Jaccard's similarity) and (B) abundance data (Bray–Curtis's similarity) for fish visual censuses made at Barú (red) and Varadero (blue) Reefs.

sponge cover), and reduction in the abundances of sea turtles, sharks and fish populations since the 1970s (*De Bakker et al., 2017*; *Jackson et al., 2014*). Reef deterioration has not been equal throughout the Caribbean with few regions still holding coral cover higher than 30% (*Gardner et al., 2003*). Most areas with relatively high coral cover are under some conservation/management program and have experienced little anthropogenic influence from land-based pollution and fisheries (*Jackson et al., 2014*). Moreover, regional and global risk assessments correlate reefs' vulnerability to their proximity to man-made stressors (*Burke et al., 2011*). The discovery of an apparently healthy reef in Varadero adjacent to the major population center of Cartagena, Colombia, apparently runs counter to the prevailing

dogma. In addition, this reef is under the influence of the Magdalena River Delta (Canal del Dique), considered as the biogeographical barrier for shallow and mesophotic coral reefs (<200 m depth), limiting the dispersion of coral larvae (*Santodomingo et al., 2013*).

The development of coral reefs under "sub-optimal" conditions (e.g., high sedimentation, nutrients) does not appear to be a widespread phenomenon, though a few disparate cases have been recently reported. These anomalous reef ecosystems can be found in warm waters (*Liddell & Ohlhorst, 1987*; *Spalding & Brown, 2015*), upwelling-influenced areas (*Bayraktarov et al., 2013*), high latitudes (*Harriot & Banks, 2002*) and naturally turbid waters (*Anthony, 2006*; *Smithers & Larcombe, 2003*). Under extreme conditions, corals have adapted and/or acclimatized to the high temperature variance, and heterotrophic feeding is their dominant feeding mode (*Teece et al., 2011*; *Hughes & Grottoli, 2013*). Most of the reefs subjected to ongoing or temporal sedimentation have growth constrains due to the limitation on light penetration. *Perry & Larcombe (2003)* predicted that reef framework development in turbid environments might be restricted or absent, limiting coral distribution to shallow waters. Correspondingly, the portions of Varadero Reef with highest coral cover are currently constrained to the shallower portions of the reef, were they appear to be autotrophic as indicated by their relatively high $Q_m$ values. Environmental conditions at Varadero Reef have changed drastically since the Spaniards arrived several centuries ago. As described by *Restrepo et al. (2017)*, before the opening of the Canal del Dique during the 16th century in the colonial period, and subsequent modifications in the 19th Century, Cartagena Bay had no river inputs and coral reefs and seagrass beds flourished inside the Bay (*Martínez et al., 2010*). The massive arrival of waters from the Magdalena River via the Canal del Dique, after the three major modifications to the channel in 1925, 1951 and 1984 (*Mogollón, 2013*), drastically changed conditions within the Bay from clear, warm-waters to a tidal estuarine environment (*Restrepo et al., 2017*). The dispersion patterns of the turbid plume of the Canal del Dique in the Cartagena Bay are highly variable depending on the hydrodynamic and meteorological conditions (*Lonin et al., 2004*). In this context, the optical properties of the water at Varadero Reef could experiment dramatic short-term changes depending on the prevailing hydro-meteorological conditions. The description of the variability in the optical properties of the water column is key to understand the energy and calcification balance of the coral community.

Varadero Reef is highly influenced by local stressors including eutrophication, agro-chemical runoff, port and industry development, and tourism activities. The main stressor being land-based pollution that flows into the Bay through the Canal del Dique (*Mogollón, 2013*). In addition to the influx of large volumes of fresh water, sediment loads arriving into the Bay can top 150 million tons per year (*Restrepo et al., 2006*). Varadero Reef appears to be a relic of the reef formations that dominated Cartagena Bay and adjacent coastal regions during the pre-Columbian period. Despite these challenging environmental conditions, our results on reef structure and species composition demonstrate that Varadero Reef is a functional ecosystem, fully developed and similar to those found on nearby reefs (e.g., Barú and Rosario Archipelago) and Caribbean reefs more broadly (*Zea, 2001*; *Claro & Cantelar-Ramos, 2003*; *Pattengill-Semmens & Semmens, 2003*; *Valderrama & Zea, 2003*; *Alvarado-Chacon, Pizarro & Sarmiento-Segura, 2011*; *Kramer, Marks & Turnbull, 2014*).

The existence of Varadero, a "paradoxical reef" (*López-Victoria, Rodríguez-Moreno & Zapata, 2015*), is a call for scientists and managers to start looking in unexpected places for similar coral reefs or carbonate reef systems as the one found at the Amazon River mouth (*Moura et al., 2016*). More importantly, Varadero may hold information on reef coral resistance, and adaptations to high sedimentation and turbidity. In this context, Varadero could serve as a natural laboratory and potentially provide source material for reseeding future reef environments. Current reef degradation challenges the initial goal of restoration ecology, meaning that returning to a pre-disturbance state might not be possible and/or practical under present climate change (*Van Oppen et al., 2017*). Tolerance to warmer and acidified waters, greater fluctuations in salinity and exposure to nutrients, herbicides and other pollutants are critical coral resilience traits. Our observations and preliminary results of ongoing research indicate that some of these traits can be found at Varadero, but further research is needed.

If the dredging for a new shipping channel is authorized by government authorities (Agencia Nacional de Licencias Ambientales—ANLA), we estimate that 25% of the reef will be directly affected and around 50% will be indirectly affected. The environmental impacts of this dredging include sediment stress (suspended and deposited), release of toxic contaminants, noise contamination, and complete destruction of benthic organisms within the dredge path (*Rogers, 1990*; *Erftemeijer et al., 2012*; *Roberts, 2012*). Depending on the intensity, duration and frequency of increased turbidity and sedimentation, the impacts on corals may include: smothering and burial, shading, bleaching, disease (*Pollock et al., 2014*), and decreased survival and recruitment success of coral larvae (*Erftemeijer et al., 2012*). Additionally, a recent review on the effect of dredging on fish suggests the potential for elevated fish mortality, especially in early life stages (eggs and larvae) (*Wenger et al., 2017*). The destruction of Varadero Reef would be a loss for the scientific community, for local stakeholders and for Colombia as a nation.

## ACKNOWLEDGEMENTS

We would like to thank the community of Bocachica, especially the Eight Brothers with whom we did all our fieldwork in Varadero and Barú. This community has welcomed and teach us about their uses of Varadero Reef and other nearby areas. Additionally, we would like to thank all the people, including the crew of Oregon State University from Terra, that have spread the word about Varadero Reef.

### Funding

This study was supported by the National Science Foundation for the project "Coral robustness: lessons from an 'improbable' reef". The funders had no role in study design, data collection and analysis, decision to publish, or preparation of the manuscript.

## Grant Disclosures

The following grant information was disclosed by the authors:
National Science Foundation.

## Competing Interests

Monica Medina and Roberto Iglesias-Prieto are Academic Editors for PeerJ. The other co-authors declare that they have no competing interests.

## Author Contributions

- Valeria Pizarro, Mateo López-Victoria, Fernando A. Zapata, Sven Zea, Roberto Iglesias-Prieto and Joseph Pollock conceived and designed the experiments, performed the experiments, analyzed the data, contributed reagents/materials/analysis tools, wrote the paper, prepared figures and/or tables, reviewed drafts of the paper.
- Sara C. Rodríguez performed the experiments, analyzed the data, contributed reagents/materials/analysis tools, prepared figures and/or tables.
- Claudia T. Galindo-Martínez performed the experiments, analyzed the data, prepared figures and/or tables.
- Mónica Medina conceived and designed the experiments, performed the experiments, analyzed the data, contributed reagents/materials/analysis tools, wrote the paper, reviewed drafts of the paper.

## Data Availability

The raw data is included as Supplemental Files.

## Supplemental Information

Supplemental information for this article can be found online at http://dx.doi.org/10.7717/peerj.4119#supplemental-information.

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
