# Peer review of "Unraveling the structure and composition of Varadero Reef, an improbable and imperiled coral reef in the Colombian Caribbean"

_PeerJ, doi:10.7717/peerj.4119_

## Round 0.1 · original submission · Major Revisions

I have heard back from two reviewers, both of whom offer constructive comments to help you improve your manuscript. One reviewer questions the use of the term "resilience" in your paper, and both offer numerous detailed points. While the comments do not necessitate a complete overhaul of the paper, there are many points to address, and thus my decision is "major revisions" are needed.
I look forward to seeing a revised version of your manuscript.

·

Basic reporting

The ms "Unraveling a resilient reef: structure and
composition of Varadero, an imperiled coral reef in
the Colombian Caribbean" describes a community of organisms on a newly disovered healthy reef in close vicinity to the outlet of the Magdalena river, Colombia.

It is unclear why the term "resilience" is used, which implies that recovery has taken place. Since the reef has just been discovered, it is unlikely that this can be established. Perhaps the authors like to suggest that the reef's good condition indicates that it has been offering "resistance" to local threats?

The report is descriptive and mainly of local importance. The discussion could be made more interesting for a wide readership by including comparisons with other areas in the region or elsewhere of reefs in the vicinity of river discharge or in murky water (e.g. Amazon river, Singapore, Jakarta Bay, etc.). Are such reefs in other areas healthy or degraded? Have healthy reefs shown resilience?

Experimental design

No comment.

Validity of the findings

No comment.

Additional comments

Additional remarks are mentioned on the ms pdf.

Reviewer 2 ·

Basic reporting

I found the manuscript to be well organized and clearly written, only finding minor grammar errors. The paper reports on the benthic (sponges, coral, and algae) and fish assemblages of a newly discovered, and apparently highly resilient coral reef inhabiting the highly polluted waters within Cartagena Bay.
Please consider the following edits:
1) Within lines 30-33, including both the total number of sponge and coral species found at Veradero Reef.
2) Within lines 51-57, authors give examples of reefs found in extreme environmental conditions, yet there is not a single example of a reef being exposed to anthropogenic stresses similar to Veradero Reef. I recommend the authors to consider Appeldoorn et al., 2015 “Mesophotic coral ecosystems under anthropogenic stress: a case study at Ponce, Puerto Rico”.

Experimental design

The experimental design as also well organized and clearly written. The survey design and statistical analyses for benthic and fish assemblages were correctly implemented.
There are several additions and modifications that I would ask the authors to consider:
1) Line 89-91 Possibly rephrasing the sentence to/similar: Data from the GPS instrument was downloaded and analyzed using the GIS software Garmin BaseCamp, from which a detailed map of the reef was subsequently produced (Figure 1).
2) Line 94 Possibly rephrasing the sentence to/similar: Annotations of coral community composition at multiple depths was analyzed as in Geister (1977).
3) Line 95 – to emphasize and not confuse the reader, please include the name of the reef. “.. a detailed profile of Veradero Reef’s coral …”
4) Line 125 – replace the word “later” with the word “downstream”

Validity of the findings

The results of the study are reasonable given the experiments performed. I congratulate the authors for including marine sponges within the surveys and analysis, they are usually left out of major and important survey efforts, even though they are an important component of any coral reef ecosystem.

Additional comments

First and foremost, I found this paper extremely interesting, especially since it highlights the discovery of a reef which seems thrive in volatile and toxic environment. As the authors state, Veradero Reef should be protected, I believe that the Colombian government and its citizens should take pride in this newly discovered coral reef, and enforce protection laws against the destruction of this resilient coral reef by the proposed dredging of the Cartagena Bay.

I do however, have several comments for the authors to take into account:
1) Line 190 – the genus Helioseris (correct) is misspelled.
2) Line 193 – the genus Undaria is no longer valid and should be replaced by Agaricia
3) Same for lines 194, 202, 207, 210, 211, 227, 232, 233,
4) Same for Figure 2 – Undaria should be changed to Agaricia
5) Figure 2 – genus Helioseris should be corrected
6) Line 225 – replace word “are” with “were” “..were also observed.”
7) Line 226 – start sentence with “In total, 38 coral species…”
8) Line 226 – after the word “identified” add the words “at this depth”
9) Line 235 – I am a confused by this sentence. The authors state that in total, there are 50 sponge species were observed at Varadero Reef, yet, there’s 38 species listed in parentheses. Maybe the authors meant to say that in total, “69” sponge species were observed?
10) Line 237-238 Proposed sentence: Survey transects at upper shallow terraces in Varadero Reef showed higher sponge species richness than that of upper shallow terraces in Baru Reef; 36 and 25 species, respectively.
11) Line 289 – see de Bakker et al. 2016 “40 Years of benthic community change on the Caribbean reefs of Curacao and Bonaire: the rise of slimy cyanobacterial mats” to possibly cite a decline in coral cover.
12) line 291 – a comma is needed within the reference after et al.
13) line 346 – I recommend switching the words “be not” to “not be”
14) Figure 4 – on the X-axis, change the word “Life” with “Live”
15) Within S1 – replace Undaria with Agaricia
16) Within S1 – Porites astreoides is misspelled
17) Within S2 – Amphimedon is misspelled

---

## Round 0.2 · Minor Revisions

As detailed by the reviewer, there are a few remaining minor details that still need attention; hence "minor revisions" is my decision. I look forward to seeing your revised manuscript.

·

Basic reporting

See previous review

Experimental design

See previous review

Validity of the findings

See previous review

Additional comments

The paper has been improved. It reads very well. The illustrations are very nice. I like the artwork in figure 2. Because the ms is almost ready for publication, I congratulate the authors.

However, I have found some minor issues for possible improvement. They are indicated in the word document.

Based on two remarks in the margin, I have explained the importance of a hyphen. This is not always a matter of personal style.

The authors did not take care of the reference list as suggested in my first review. In several references, the spelling of the names of authors is incorrect. In many references, the names of authors still lack initials. I have marked some examples.

I have also checked the supplementary material. I suggest that taxon names in tables should be presented in alphabetical order.

---

## Round 0.3 · accepted · Accept

The manuscript is well-revised, and is now acceptable for publication. Make sure you do the following small edit at the proof stage or earlier:
1. line 242: "zoanthids" to "zoantharians".

I look forward to seeing your published paper!